# Trousseau’s Syndrome and Marantic Endocarditis in a Patient with Pulmonary Adenocarcinoma: A Case Report and a Brief Review of the Literature

**DOI:** 10.3390/reports8040215

**Published:** 2025-10-27

**Authors:** Leandro Cosco, Margherita Padeletti, Andrea Sorrentino, Massimo Milli, Rossella Marcucci

**Affiliations:** 1Department of Experimental and Clinical Medicine, University of Florence, Azienda Ospedaliero-Universitaria Careggi, 50134 Florence, Italy; andrea.sorrentino@unifi.it (A.S.); rossella.marcucci@unifi.it (R.M.); 2Cardiology and Electrophysiology Unit, Santa Maria Nuova Hospital, USL Toscana Centro, 50122 Florence, Italy; margherita.padeletti@uslcentro.toscana.it (M.P.); massimo.milli@uslcentro.toscana.it (M.M.)

**Keywords:** Trousseau’s syndrome, venous thromboembolism, marantic endocarditis, cancer

## Abstract

**Background and Clinical Significance:** Trousseau’s syndrome, characterized by recurrent thromboembolic events and non-bacterial thrombotic endocarditis, represents a severe paraneoplastic condition associated with poor prognosis in cancer patients. Due to the growing life expectancy of cancer patients, Trousseau’s syndrome is becoming more frequent. Consequently, risk of thrombosis and bleeding assessment, as well as early diagnosis and opportune therapy will gain importance. **Case Presentation:** We describe a case of a 63-year-old Caucasian male presenting with ischemic stroke. During management, he developed a mitral valve marantic endocarditis, and finally the diagnosis of pulmonary adenocarcinoma was performed. The case description is followed by a brief review of the relevant literature on the condition. **Discussion and Conclusions**: This case highlights the complexity of diagnosing and managing Trousseau’s syndrome. Early recognition, appropriate anticoagulation strategies, and the need for multidisciplinary management are crucial to improve the outcomes and the quality of life for cancer patients.

## 1. Introduction and Clinical Significance

Thromboembolic events represent the second leading cause of death among patients with malignant neoplasms [1] and may even constitute the first clinical manifestation of an underlying cancer [2]. The combination of recurrent venous and/or arterial thromboses in association with non-bacterial (marantic) endocarditis is known as Trousseau’s syndrome. This condition was first described in 1865 by Armand Trousseau [3], who self-diagnosed it two years later, shortly before his death due to gastric cancer [4]. Trousseau’s syndrome is associated with poor prognosis and reduced quality of life in cancer patients [5]. However, specific histological subtypes and targeted oncologic therapies may influence clinical outcomes [5] making early recognition and diagnosis essential. In addition, although it is a well-known condition, thrombotic and bleeding risk assessment, as well as diagnostic and therapeutic strategies are still under debate. Here we report the case of a 63-year-old man diagnosed with Trousseau’s syndrome, followed by a brief review of the current literature on this condition. The aim of this paper is to raise awareness of the clinical manifestations of this severe condition, enabling clinicians to achieve early diagnosis and initiate timely treatment.

## 2. Case Presentation

A 63-year-old Caucasian male patient presented to our Emergency department (ED) with dysphagia and dysarthria, which had begun approximately seven hours earlier. Upon arrival, he was alert, aphasic, with a right-sided facial nerve deficit, and intact Mingazzini signs. The National Institutes of Health Stroke Scale (NIHSS) score was 6, while the modified Rankin Scale score was 3. Cardiac auscultation revealed a regular rhythm with a 3/6 systolic murmur. The remainder of the physical examination was unremarkable.

At admission, the patient’s body weight was 75 kg, with a body mass index (BMI) of 26 kg/m^2^. Laboratory tests (Table 1) showed a white blood cell (WBC) count of 9670/μL (reference range: 4000–10,000/μL), hemoglobin of 11 g/dL (13–17 g/dL), platelet count of 170,000/μL (150,000–400,000/μL), markedly elevated D-dimer at 21,268 EEU (0–500 EEU), serum creatinine of 0.76 mg/dL (0.6–1.2 mg/dL), and a negative C-reactive protein (CRP) test (<5 mg/L). Autoimmune markers including antinuclear antibodies (ANA), extractable nuclear antigens (ENA), anticardiolipin antibodies, and β2-microglobulin were negative. Admission blood cultures were negative.

Past medical history included hypertension, hyperuricemia, and follicular lymphoma, initially treated with the CHOP + bendamustine regimen and later with rituximab. One month earlier, enlarged lymph nodes with poorly defined margins raised suspicion for disease recurrence, but treatment was deferred due to the absence of GELF criteria [6]. Two weeks before the presentation, the patient developed left femoropopliteal deep vein thrombosis and started edoxaban 60 mg daily. His other chronic medications included ramipril 5 mg and allopurinol 300 mg. The patient was also a heavy smoker (35 pack-years).

A cranial CT scan and CT angiography revealed ischemic lesions in the left post-central insular gyrus and the left occipital gyrus, along with a complete occlusion of the left middle cerebral artery (Figure 1). No indication for endovascular treatment was found. Follow-up cranial CT scans excluded hemorrhagic infarctions. Anticoagulation with edoxaban was continued. No arrhythmias were detected on Holter ECG.

Transthoracic echocardiography (TTE) showed normal left ventricular systolic function with severe mitral regurgitation but no evidence of endocavitary masses. A transesophageal echocardiogram (TEE) ruled out a patent foramen ovale and identified endocarditis of the mitral valve, with distal thickening of both mitral leaflets (maximum 3 mm) resulting in multiple, predominantly eccentric, regurgitant jets, causing severe mitral regurgitation (Figure 2), with no indications for urgent/emergency cardiac surgery. In addition, the patient, who was still dysphagic, developed cough, dyspnea, and oxygen desaturation, leading us to diagnose an intercurrent aspiration pneumonia. Considering findings suggestive of endocarditis (culture-negative) and coexisting aspiration pneumonia, empirical antibiotic therapy was initiated with ceftriaxone, vancomycin, and clarithromycin. However, the patient remained afebrile, with negative blood cultures, WBC counts of 6390/μL, and stable CRP levels between 10 and 12 mg/L. Serial procalcitonin (PCT) levels were always negative. Furthermore, serological testing for pathogens typically associated with culture-negative endocarditis, particularly *Coxiella burnetii*, *Bartonella*, and *Brucella*, yielded negative results.

Therefore, given the negative blood cultures, serological tests, and absence of fever or other minor criteria, the patient did not meet the Duke criteria for infective endocarditis (IE) [7]. Suspecting lymphoma recurrence as reason for the endocarditis, a CT scan of the neck, chest, and abdomen was performed. The examination revealed ischemic-infarct areas in the spleen and a tissue mass with contrast enhancement in the left hilar para-aortic region causing narrowing of the bronchial structures (Figure 3). Consequently, bronchoscopy was performed with biopsy of the left upper lobe bronchus.

Soon after, the patient developed thrombosis of the proximal left axillary artery and right axillary vein at a PICC (Peripherally Inserted Central Catheters) line site. Edoxaban was replaced by LMWH at an anticoagulant dosage. The pulmonary biopsy report revealed the presence of pulmonary adenocarcinoma. Consequently, the mitral endocarditis was categorized as marantic, and its treatment with cardiac surgery was ruled out due to the high operative risk and the patient’s overall frailty. Due to the patient’s rapid clinical deterioration and death within two months of diagnosis, the results of further molecular analyses (EGFR, ALK, ROS1, etc.) were not available at the time of death, while PD-L1 testing had already been performed and was negative.

## 3. Discussion

According to the ESC cardio-oncology guidelines [1], malignancies increase the risk of venous thromboembolism (VTE) by fivefold and the risk of arterial thromboembolism by twofold. In addition, the life expectancy of cancer patients has increased in recent decades, partly due to new anti-cancer therapies and early diagnosis. Therefore, it is safe to assume that the burden of cancer-associated thrombosis (CAT) will become greater, and the prevention and treatment of this condition will gain relevance.

In the treatment and secondary prevention of venous thromboembolism in cancer patients, the guidelines recommend LMWH. LMWH is generally preferred to unfractionated heparin (UFH) because LMWH does not require blood test monitoring and hospitalization. LMWH was also more effective than warfarin in reducing the risk of VTE without increasing the risk of bleeding in patients with high thrombotic risk [8,9]. Direct acting oral anticoagulants (DOACs) such as edoxaban, rivaroxaban, or apixaban [10,11,12] are potential alternatives (Table 2).

In head-to-head trials with LMWH, all three DOACs were non-inferior in treating venous thromboembolism; however, only apixaban, in the CARAVAGGIO trial, did not increase the bleeding risk. Consequently, the 2023 update of the ASCO guidelines [13] included apixaban among the preferred DOACs. In the API-CAT trial [14], reduced-dose apixaban (2.5 mg twice daily) was found to be non-inferior to the full-dose regimen (5 mg twice daily) in preventing venous thromboembolism in cancer patients who had already completed 6 months of anticoagulant therapy for VTE/PE (pulmonary embolism). Moreover, the reduced dose was associated with a lower rate of clinically relevant bleeding compared to the full-dose regimen.

**Table 2 reports-08-00215-t002:** Anticoagulation acute strategies in cancer patients with thromboembolism [1,13,15].

Therapy	Indications	Advantages	Limitations/Contraindications
Low Molecular Weight Heparin (LMWH)	First-line for treatment and prevention of VTE in cancer patients.	Proven efficacy, reduced VTE vs. warfarin, easy dosing	Injection route, bleeding risk, impaired renal function.
Direct Oral Anticoagulants (DOACs) (edoxaban, rivaroxaban, apixaban)	Alternative to LMWH in selected patients.	Oral administration, non-inferior efficacy. Apixaban does not increase bleeding risk.	Avoid in unresected GI/GU cancers, CrCl < 15, platelets < 50 k, recent surgery, bleeding risk
Vitamin K Antagonists (VKAs)	Alternative if DOACs/LMWH not suitable	Oral, long experience	Drug–food interactions, INR monitoring required, less preferred
Unfractionated Heparin (UFH)	Hospitalized patients, rapid reversal needed	Short half-life, reversible	Requires monitoring (aPTT), IV route
Anticoagulation Duration	≥6 months recommended (individualized)	Reduces recurrence	Reassess bleeding periodically, especially in advanced cancer

Nevertheless, DOACs are contraindicated in patients with one of the following bleeding factors: unoperated gastrointestinal (GI) or genitourinary (GU) malignancies, history of recent bleeding or within 7 days of a major surgery, significant thrombocytopaenia (platelet count < 50,000/μL), severe renal dysfunction (creatinine clearance (CrCl) < 15 mL/min), or GI comorbidities. The minimal duration of anticoagulation is 6 months [1,15]. However, patients with cancer are also at high risk of bleeding during anticoagulant treatment. For this reason, a periodic assessment of the risk/benefit ratio should be performed. The most famous and well validated score to calculate VTE risk in cancer patients is the Khorana score [16], while existing risk scores for bleeding perform poorly after CAT [17].

Although edoxaban was an appropriate therapeutic choice, our patient developed thrombosis despite treatment, which, together with the concomitant finding of marantic endocarditis, led us to switch to LMWH. This apparent treatment failure was likely attributable to the high prothrombotic potential of lung adenocarcinoma. It is mandatory to highlight that the patient described in the report came to our attention before the publication of the 2022 ESC cardio-oncology guidelines and the 2023 update of the ASCO guidelines. He developed episodes of both venous and arterial thrombotic non-infective (marantic) endocarditis with multiple embolic events involving the brain and spleen. Recurrent thrombosis with no other explanation, associated with marantic endocarditis, in a cancer patient is known as Trousseau’s syndrome [4]. Although neither the ASCO guidelines nor the ESC cardio-oncology guidelines specifically mention it, Trousseau’s syndrome is associated with reduced quality of life and worsened prognosis [5]. In a recent cohort study by Wan H. et al., the overall incidence in hospitalized cancer patients was 8.0 cases per 1000 person-years (p-y), with the highest incidence in the first year after cancer diagnosis (15.0 cases per 1000 p-y), decreasing to 6.3 cases per 1000 p-y in the second year, and 4.2 cases per 1000 p-y thereafter [18]. A study by Sørensen et al. found that the one-year survival rate for the cancer group with VTE was 12%, compared with 36% in the control group without VTE [19]. In the ONCOTHROMB-01 [20] cohort study, 416 patients with gastrointestinal (pancreatic, gastric, esophageal, and colorectal) and non-small cell lung cancers were enrolled. VTE was diagnosed concomitantly with neoplasm in 30 patients. After 18 months of follow-up, only 30% of patients with CAT were alive, compared with 70% of patients without CAT. At 18 months, pancreatic cancer was the malignancy most strongly associated with CAT incidence, followed by lung cancer. The highest incidence peak was observed between the third and sixth months of follow-up), in line with the previous literature reporting that most VTEs occur within the first year after the initial clinical manifestations of cancer [18].

This paraneoplastic syndrome involves hypercoagulability related to the patient, cancer, and treatment (Table 3). Patient-related factors include age, inactivity, and cardiovascular comorbidities. Cancer-related factors include tissue factor (TF), plasminogen activator inhibitor (PAI-1), mucins, cytokines, and hypoxia [4]. TF directly induces the conversion of factor VII to factor VIIa, resulting in the constitutive activation of the coagulation cascade. It has been reported that, in addition to coagulation, TF is also associated with cancer metastasis and angiogenesis [21]. Moreover, the sialic acid moieties of mucin from adenocarcinomas cause a nonenzymatic activation of factor X. All in all, hypoxia (decreased oxygenation) could increase the expression of genes that facilitate coagulation, including TF and PAI-1 [4]. PAI-1 inhibits the activation of plasminogen, a key step in the dissolution of blood clots. On the other hand, about treatment related factors, chemotherapeutic agents such as platinum compounds, hormonal agents, tamoxifen, growth factors (granulocyte colony-stimulating factor and erythropoiesis-stimulating agents) and antiangiogenic agents increase the risk of thrombosis [21]. Chemotherapy itself, independently of the underlying pathophysiology of the neoplastic process, induces a hypercoagulable state by also acting on TF and PAI-1. Indeed, Wrzeszcz et al. demonstrated that adjuvant therapy in patients with invasive breast cancer significantly increased plasma concentrations of TF and PAI-1 [22]. However, when anticoagulant therapy is added, the hemostatic balance may shift toward bleeding risk. In fact, several studies have shown that the concomitant administration of DOACs and chemotherapeutic agents increases the risk of bleeding, primarily due to pharmacokinetic interactions involving CYP3A4 and P-glycoprotein (P-gp) inhibitors or inducers. The tyrosine kinase inhibitors (TKIs) represent the most clinically significant class of agents in terms of drug–drug interactions (DDIs) with DOACs, as they are potent inhibitors of P-gp [23].

Nowadays CVCs (Central Venous Catheters) are gaining a pivotal role for the long-term administration of anti-cancer drugs and blood sampling, but they are associated with an increased thrombotic risk [24], as illustrated by the case presented. Several types of cancer are associated with Trousseau’s syndrome. Of these, lung cancer is most frequently associated with malignancy-related ischemic stroke, which is the most common presentation [25]. Although the patient’s history of follicular lymphoma represents a potential confounding factor, the absence of GELF criteria suggests limited biological activity at that time. In addition, while follicular lymphomas may be associated with a modest increased risk of venous and arterial thromboembolism [26], the lung adenocarcinoma is strongly associated with a prothrombotic state, through mechanisms including mucin secretion and activation of coagulation pathways [4,27]. For these reasons, albeit a contribution from the lymphoma cannot be entirely excluded, we considered lung adenocarcinoma to be the most plausible cause of the patient’s thromboembolic manifestations.

Even though the diagnosis of Trousseau’s syndrome is generally associated with a worse prognosis in such patients, early recognition—particularly with regard to histological subtypes—may improve clinical outcomes. In a recent small study, Yoshimine et al. [5] observed that patients with more favorable prognoses exhibited non-adenocarcinoma histotypes, a high frequency of EGFR mutations, and were more likely to receive immune checkpoint inhibitor (ICI) therapy. Based on these observations, the authors propose that when Trousseau’s syndrome develops in patients with lung cancer, continuous heparin therapy should be initiated, and genetic testing along with PD-L1 immunostaining should be promptly performed to guide appropriate treatment. Unfortunately, this study has several limitations, including its monocentric design and limited sample size. Therefore, this single study is insufficient, and further multicenter studies with larger patient cohorts will be necessary.

Additionally, since our patient did not meet the Duke criteria and had an active tumor, the endocarditis seen on TEE was categorized as marantic endocarditis [7], typically linked to paraneoplastic hypercoagulability from solid tumors, especially lung, gastrointestinal, and pancreatic cancers. Its highest incidence is typically observed in individuals aged between the fourth and eighth decades of their life [27]. Differentiating the non-bacterial thrombotic endocarditis (NBTE) vegetations from IE vegetations can be extremely challenging. In our case, several clinical elements strongly supported the diagnosis of NBTE, including the presence of lung adenocarcinoma, stroke, and valvular thickening—all consistent with the clinical profile of NBTE, together with the absence of fever, leukocytosis, bacteremia, and typical immunologic or vascular phenomena of IE. This form of endocarditis is often underestimated on TTE but should be suspected in the presence of Trousseau’s syndrome. Indeed, TTE is less sensitive than TEE, particularly for detecting vegetations smaller than 5 mm that are often associated with NBTE [27,28]. The mitral valve is the most frequently affected. Its involvement is associated with a high rate of clinical complications and mortality [29]. There have been instances of improvement or resolution of NBTE vegetations with the treatment of the underlying disease, which is often neoplastic, but sometimes also autoimmune or viral (e.g., SARS-CoV-2) [27]. Regarding anticoagulation, the ESC guidelines recommend LMWH as the first-line treatment [7] in marantic endocarditis. Valid alternatives may include vitamin K antagonists (VKA) or UFH. There are no data to support the use of direct oral anticoagulants in NBTE [30]. The duration of the anticoagulation therapy is not well established, because large clinical trials are lacking. The role of surgery is controversial, but it may be considered in select cases with significant persistent valvular dysfunction [31]. Antibiotics are not indicated.

## 4. Conclusions and Take-Home Messages

Our case highlights the severity of Trousseau’s syndrome and marantic endocarditis, conditions that should be suspected in cancer patients. The life expectancy of these patients will increase, as well as the incidence of thrombotic complications associated with cancer. Therefore, the prevention and treatment with anticoagulants of this condition will gain relevance, in order to significantly reduce its morbidity and mortality. However, the risk of bleeding must be weighed, especially in advanced cancer patients, necessitating a multidisciplinary approach and a stretched follow-up involving an expert in thrombosis and hemostasis.

## Figures and Tables

**Figure 1 reports-08-00215-f001:**
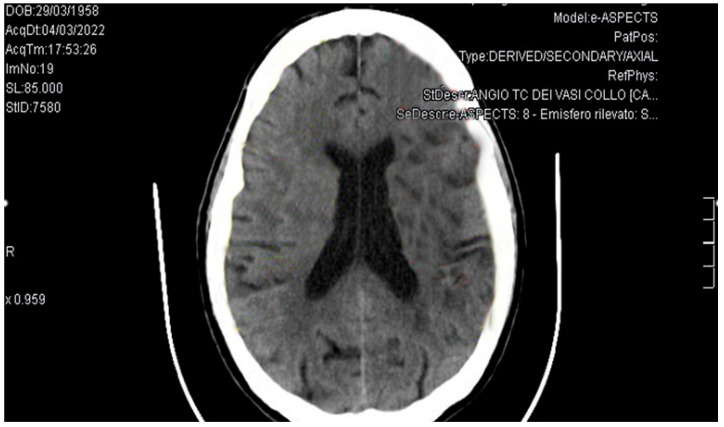
Cranial TC scan: ischemic area in the left post-central insular gyrus and the left occipital gyrus.

**Figure 2 reports-08-00215-f002:**
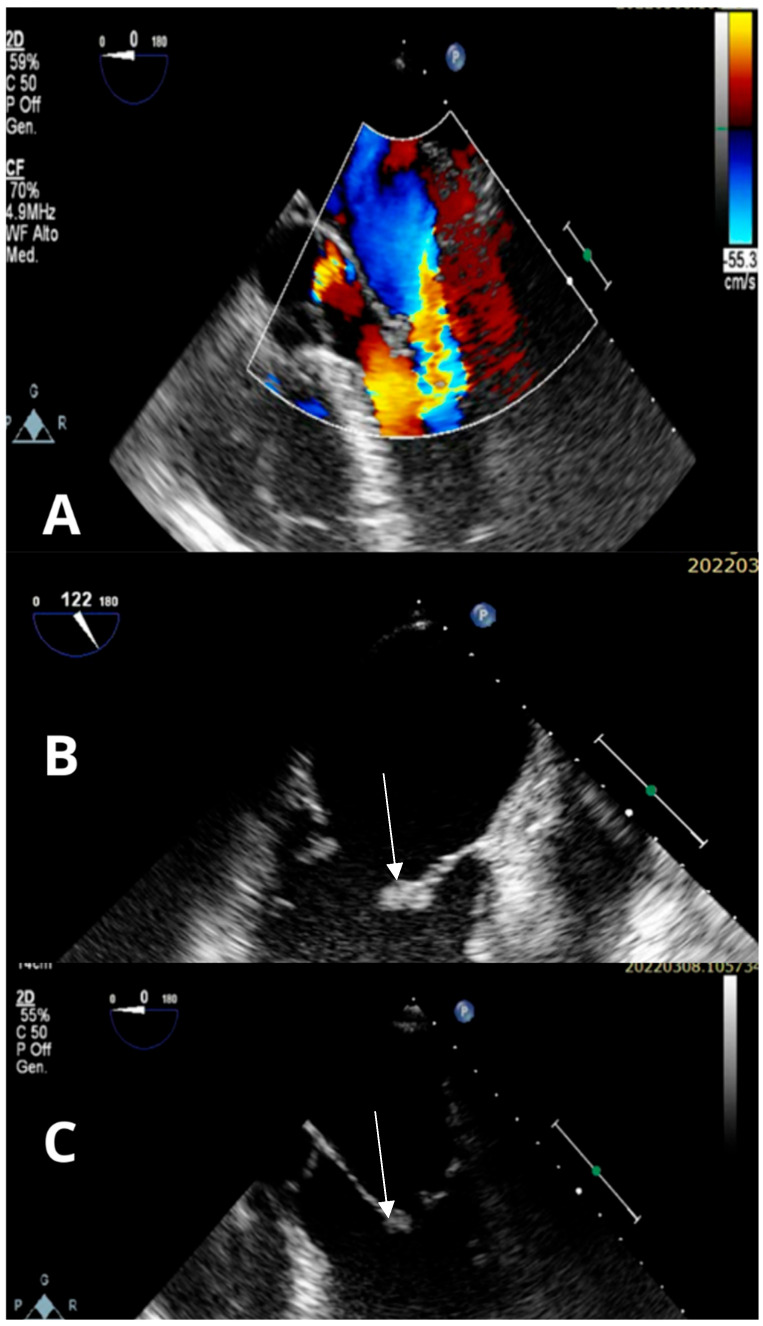
(**A**–**C**): TEE: Distal thickening of both mitral leaflets (white arrows), with a maximum thickness of three millimeters, impairing proper systolic coaptation and resulting in multiple, predominantly eccentric, regurgitant jets, overall consistent with severe mitral regurgitation. Findings suggestive of mitral valve endocarditis.

**Figure 3 reports-08-00215-f003:**
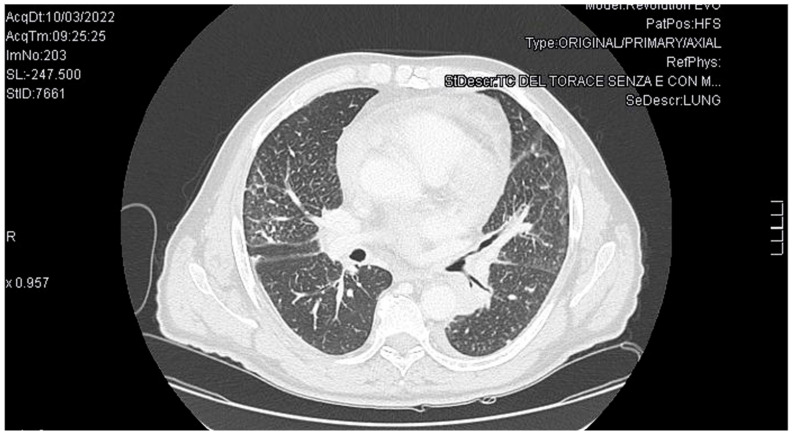
Chest CT scan: a tissue mass in the left hilar para-aortic region causing narrowing of the bronchial structures.

**Table 1 reports-08-00215-t001:** Blood tests at admission with normal values.

Parameter	Result at Admission	Reference Range
**WBC count**	9670/μL	4000–10,000/μL
**Hemoglobin**	11 g/dL	13–17 g/dL
**Platelet count**	170,000/μL	150,000–400,000/μL
**D-dimer**	21,268 EEU	0–500 EEU
**Serum creatinine**	0.76 mg/dL	0.6–1.2 mg/dL
**CRP**	Negative	Negative
**ANA, ENA, Anticardiolipin antibodies, β2-microglobulin**	Negative	Negative

**Table 3 reports-08-00215-t003:** Cancer-related thromboembolic risk factor [4,18].

Category	Risk Factors
Patient-related	Older age, immobility, comorbidities (e.g., hypertension), history of thrombosis
Tumor-related	Histological type (especially adenocarcinoma), tumor burden, metastasis
Biological mediators	Tissue factor (TF), mucins, PAI-1, cytokines, hypoxia
Treatment-related	Chemotherapy (e.g., platinum compounds), hormonal therapy, antiangiogenics
Drug interactions	DOAC metabolism affected by CYP3A4/P-gp inhibitors (e.g., tyrosine kinase inhibitors)
Procedural	Central venous catheters (CVCs), recent surgery

## Data Availability

The original contributions presented in this study are included in the article. Further inquiries can be directed to the corresponding author.

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
