# Peer review of "Trousseau’s Syndrome and Marantic Endocarditis in a Patient with Pulmonary Adenocarcinoma: A Case Report and a Brief Review of the Literature"

_reports, 2025, doi:10.3390/reports8040215_

Round 1
Reviewer 1 Report
Comments and Suggestions for Authors
The article "Trousseau's syndrome and marantic endocarditis in a patient with pulmonary adenocarcinoma: a case report and review of literature" describes a case of a man with ischemic stroke and marantic endocarditis of the mitral valve, who was diagnosed with lung adenocarcinoma. The title of the article fully reflects the content of the article.
In the Abstract section, the authors indicated that Trousseau syndrome is a severe paraneoplastic condition and is associated with an unfavorable prognosis in cancer patients. In the section, the authors briefly presented a 63-year-old man with ischemic stroke and mitral valve marantic endocarditis, who was eventually diagnosed with lung adenocarcinoma. It is important that in addition to the description of the clinical case, the article presents a review of the current literature on this disease. The conclusion that the authors presented at the end of the article is important for clinical practice.
The "Keywords" presented in the article correspond to the content of the article and are necessary.
In section "1. Introduction" the authors briefly described thromboembolic complication in patients with malignant diseases as. In the section the authors indicated that specific histological subtypes and targeted oncological therapy may affect clinical outcomes. Thus, the study of various clinical cases associated with ischemic stroke and mitral valve marantic endocarditis is important for early diagnosis and change of therapy in order to combat neoplastic processes. In this section it is recommended to indicate the purpose of the study.
The authors then sequentially presented sections such as "2. Ethical Statement and Patient Consent" and "3. Clinical Case". The description of the clinical case is accompanied by drawings that are important for the diagnosis and prognosis of the disease, as well as understanding and analyzing the results of the patient's examination.
In section "4. Discussion" the authors analyze their results, for which they involved published data from other research groups. A significant part of the section is devoted to the description of known examples of treatment and secondary prevention of venous thromboembolism in cancer patients. The description of Trousseau syndrome associated with various cancers is interesting and important. Of all cancers, lung cancer is most often associated with ischemic stroke caused by malignant neoplasms. The analysis of published studies allowed the authors of this article to propose starting permanent therapy in patients with lung cancer with heparin when Trousseau syndrome develops, as well as immediately conducting genetic testing with PD-L1 immunostaining to determine adequate therapy. It is important that the authors pointed out a number of limitations of this study, such as a monocentric design and limited sample size. This must be overcome. The section contains tables that are necessary for understanding the study results.
I agree with the conclusions that the authors presented in section "5. Conclusions". The conclusions are that early diagnosis and adequate anticoagulant therapy can significantly reduce mortality and morbidity in cancer patients suspected of having Trousseau syndrome and marantic endocarditis. It is important to consider the risk of bleeding.
The article is important for clinical practice. The text of the article is written clearly. The manuscript did not raise any ethical issues. All references to publications in the References section are necessary and correct, and are written in the correct style. I have no concerns about the similarity of this article to other articles published by the same authors. Competing interests of the authors do not create bias in the presentation of results and conclusions.
Reviewer 2 Report
Comments and Suggestions for Authors
The abstract and the title represent the content of the article.
The introduction is short but suffices for the framing the problem.
The case is presented clearly except for two issues. First: what were the clinical signs to suspect pneumonia? Second: the yellow and red lines on the CT of the brain should be removed.
The conclusion could serve as take-home message and could be labeled as such
Reviewer 3 Report
Comments and Suggestions for Authors
The patient had a history of follicular lymphoma and was recently treated with edoxaban, making it difficult to determine if the episodes were solely attributable to lung cancer-associated Trousseau’s syndrome. Nevertheless, certain seminal or extensive cohort studies regarding cancer-associated thrombosis and marantic endocarditis remain unaddressed. Furthermore, the authors determine marantic endocarditis; nonetheless, the case does not fully satisfy Duke’s criteria for infective endocarditis. Even if they support NBTE, culture-negative infective endocarditis cannot be completely ruled out. The research emphasizes the preference for LMWH but fails to examine possible alternative treatments, such as surgical interventions for valve failure and genetic testing for lung cancer. There were no new ideas about how to manage things that were talked about. In my perspective, the biggest problem with the case is that it doesn't contribute much new information and is made weaker by confusing elements, a shallow literature study, and conflicting ethical pronouncements.
Comments on the Quality of English LanguageThe quality of the English language is understandable, and I was able to follow the meaning of the manuscript.
Reviewer 4 Report
Comments and Suggestions for Authors
The case is clinically interesting; however, there are some comments to improve the manuscript.
The abstract is very deficient. There is no data about the presented case and consequently the conclusion is not related to the content of the abstract.
The introduction is very short with no knowledge gap or what new this case can add.
The ethical statement can be moved to the declarations section.
Remove any dates in the case description as these are identifiers.
Follow CARE guidelines in describing the case.
The patient developed thrombosis on anticoagulation, which may indicate treatment failure or inadequate treatment. Discuss this further.
The rationale for AB in non-febrile and culture-negative patients should be stated (ex, culture-negative IE).
Add tables with labs and normal values.
The discussion is accepted; however, this is not a literature review. Either perform systematic literature review or report the case and discuss relevant literature only.
Round 2
Reviewer 3 Report
Comments and Suggestions for Authors
I have already decided to reject this manuscript and see no reason to change my decision. The patient’s history of follicular lymphoma and recent edoxaban use make it difficult to attribute the thromboembolic episodes solely to lung cancer–associated Trousseau’s syndrome. Several seminal and large-scale studies on cancer-associated thrombosis and marantic endocarditis were not addressed. Although the authors conclude that the patient had marantic endocarditis, the case does not fulfill Duke’s criteria for infective endocarditis, and culture-negative endocarditis cannot be excluded. The discussion focuses narrowly on LMWH while neglecting alternative management strategies, such as surgical valve intervention or genetic evaluation for lung cancer. Overall, the manuscript lacks novelty, contains interpretive ambiguities and insufficient literature context, and provides little contribution to current clinical understanding
Comments on the Quality of English LanguageThe manuscript is written in generally understandable English; however, the language requires improvement for clarity and precision.